# High-performance flexible p-type Ce-filled Fe$_3$CoSb$_{12}$ skutterudite thin film for medium-to-high-temperature applications

Dou Li[1,4], Xiao-Lei Shi[2,4], Jiaxi Zhu[1], Tianyi Cao[2], Xiao Ma[1], Meng Li[2], Zhuokun Han[3], Zhenyu Feng[1], Yixing Chen[1], Jianyuan Wang[3], Wei-Di Liu[2], Hong Zhong[1] ✉, Shuangming Li [1] ✉ & Zhi-Gang Chen [2] ✉

P-type Fe$_3$CoSb$_{12}$-based skutterudite thin films are successfully fabricated, exhibiting high thermoelectric performance, stability, and flexibility at medium-to-high temperatures, based on preparing custom target materials and employing advanced pulsed laser deposition techniques to address the bonding challenge between the thin films and high-temperature flexible polyimide substrates. Through the optimization of fabrication processing and nominal doping concentration of Ce, the thin films show a power factor of >100 µW m$^{-1}$ K$^{-2}$ and a $ZT$ close to 0.6 at 653 K. After >2000 bending cycle tests at a radius of 4 mm, only a 6 % change in resistivity can be observed. Additionally, the assembled p-type Fe$_3$CoSb$_{12}$-based flexible device exhibits a power density of 135.7 µW cm$^{-2}$ under a temperature difference of 100 K with the hot side at 623 K. This work fills a gap in the realization of flexible thermoelectric devices in the medium-to-high-temperature range and holds significant practical application value.

The increasing demand for wearable electronic devices underscores the need for stable, efficient, and flexible energy solutions to meet their power requirements[1]. Compared to traditional batteries[2], flexible thin-film-based thermoelectric materials and devices exhibit characteristics such as lightweight, bendability, wearability, and simple structure[3]. They can closely adhere to heat sources, converting temperature differences into electrical energy, serving as power sources and therefore greatly reducing or even eliminating reliance on external power sources[4]. To achieve high energy conversion efficiency, the thin-film materials constituting these devices should possess excellent thermoelectric performance. The figure of merit $ZT = (S^2\sigma/\kappa)T$ of thermoelectric materials is a parameter that measures the performance of the material in the thermoelectric conversion, where $S^2\sigma$ represents the power factor consisting of electrical conductivity $\sigma$ and the Seebeck coefficient $S$, $\kappa$ is the thermal conductivity contributed by

both electrons ($\kappa_e$) and lattice ($\kappa_l$) components ($\kappa = \kappa_e + \kappa_l$), and $T$ is the absolute temperature (K)[5]. However, optimizing the $ZT$ value of thermoelectric materials has always been challenging due to the close correlation of many physical parameters, including $S$, $\sigma$, and $\kappa_e$, with the carrier concentration $n$[6]. Till now, band engineering[7], defect engineering[8], and other strategies have been commonly employed for structural and compositional control to optimize overall thermoelectric performance[9].

In addition to requiring excellent thermoelectric performance, high stability and flexibility are also important considerations for practical applications[10]. Currently, flexible thermoelectric thin films are mainly divided into three types, namely organic, inorganic, and composite films[11]. Among them, organic films exhibit advantages such as high flexibility, low cost, low toxicity, and lightweight[12]. However, their lower thermoelectric performance, especially the difficulty in

[1]State Key Laboratory of Solidification Processing, Northwestern Polytechnical University, Xi'an 710072, P. R. China. [2]School of Chemistry and Physics, ARC Research Hub in Zero-emission Power Generation for Carbon Neutrality, and Centre for Materials Science, Queensland University of Technology, Brisbane, Queensland 4000, Australia. [3]MOE Key Laboratory of Material Physics and Chemistry Under Extraordinary Conditions, School of Physical Science and Technology, Northwestern Polytechnical University, Xi'an 710072, P. R. China. [4]These authors contributed equally: Dou Li, Xiao-Lei Shi. ✉e-mail: zhonghong123@nwpu.edu.cn; lsm@nwpu.edu.cn; zhigang.chen@qut.edu.au

improving the $S$, hinders their commercialization[12]. Also, their poor temperature resistance, which is usually used at <500 K, also limits their application in medium-to-high-temperature environments[12]. Although the thermoelectric properties of organic-inorganic composite films have risen to some extent[10], the mechanism behind performance enhancement is complex[12], and their temperature resistance is still limited by the organic matrix[10]. Therefore, depositing inorganic materials on flexible substrates is another way of achieving flexible thermoelectric thin films. Among inorganic thermoelectric thin films, $Bi_2Te_3$[13,14], as well as silver chalcogenides such as $Ag_2Se$[15,16], have exhibited excellent near-room-temperature $ZT$ values ($ZT$ > 1 at 400 K). However, the reported inorganic flexible thermoelectric thin films are currently limited to functionalities near room temperature[10]. This is because with increasing the temperature, the thermoelectric performance of these inorganic flexible films gradually deteriorates in the medium-to-high temperature range (*e.g.*, from 500 K to 700 K)[15,17], and Te and Se are prone to volatilize in this temperature range, leading to reduced film stability[18]. Considering the enormous application potential and commercial value of flexible thermoelectric materials and devices designed for higher temperatures, especially in nonplanar thermoelectric power generation and refrigeration, there is an urgent need for a new type of flexible film. This film must possess high thermoelectric performance at higher temperatures with outstanding flexibility and stability, which is still a considerable challenge that needs to be tackled.

Compounds with the skutterudite structures have garnered widespread concern for their excellent thermal stability, appropriate $S$, and enhanced $\sigma$[19–23], achieving preliminary success in the preparation of thermoelectric thin-film materials for higher-temperature applications[24]. In the reported skutterudite thin films, most are currently rigid films due to the long-standing challenge of compatibility between skutterudite and flexible organic substrates. Currently, n-type $CoSb_3$-based skutterudite thin films have shown significant progress, with reported $S^2\sigma$ of 210 μW m$^{-1}$ K$^{-2}$ at room temperature[25] and $ZT$ of 1.1 at 683 K through rational elemental doping[26]. However, $CoSb_3$-based thin films still face two major challenges, namely realizing highly flexible films (*i.e.*, solving the compatibility issue between skutterudite thin film and high-temperature flexible organic substrates), and achieving high-performance p-type films. For p-type skutterudite thin films, the low $\sigma$ and $S$ result in less-than-ideal thermoelectric performance, as summarized in Supplementary Table 1[24,27–37]. Additionally, the brittleness of inorganic skutterudite materials limits their practical application in irregular bending structures. Therefore, developing high-performance p-type flexible skutterudite thin films is of utmost importance.

To address this long-standing challenge, through the guidance of the first-principles calculations and fabrication optimization, we successfully achieve p-type $Fe_3CoSb_{12}$-based skutterudite thin films with exceptional stability, flexibility, and thermoelectric properties in the medium-to-high temperature range. We address the bonding issue between thin films and high-temperature flexible polyimide (PI) by developing self-made target materials and employing advanced pulsed laser deposition (PLD) techniques. By adjusting the nominal doping concentration of Ce, we optimize the hole carrier density, achieving a high $S^2\sigma$ exceeding 100 μW m$^{-1}$ K$^{-2}$, resulting in a $ZT$ value approaching 0.6 at 653 K. Furthermore, our flexible films show outstanding flexibility with only a 6 % change in electrical transport properties after over 2000 bending cycle tests at a radius $r$ of 4 mm, indicating strong adhesion between the film and the high-temperature flexible substrate. Additionally, we introduce, for the first time, a p-type $Fe_3CoSb_{12}$-based flexible device, achieving a power density $\omega$ of 135.7 μW cm$^{-2}$ at a temperature difference ($\Delta T$) of 100 K with the hot end at 623 K. This groundbreaking work fills a critical gap in the realization of flexible thermoelectric devices in the medium-to-high temperature range, holding significant practical application value.

## Results

To achieve p-type $CoSb_3$-based thin films, one should first design the nominal composition. It is well known that Fe has one fewer 3d electron than Co, substituting Fe on Co sites in the $Co_4Sb_{12}$ structure can generate one hole in the valence band[38,39]. Therefore, substituting Fe for Co can induce to formation of p-type skutterudite. However, Fe-doping brings about severe instability in skutterudite phase formation due to charge imbalance[40–42]. Hence, it is necessary to fill the high charge state filler atoms to neutralize the high hole concentration and achieve electrical neutrality conditions in the p-type skutterudite system. Ce has been proven to be an excellent filler for skutterudite due to its outstanding performance and low cost in bulk materials. Therefore, the synergistic control of Ce-filling and Fe-substitution for Co aims to regulate the p-type thermoelectric performance of the pristine $CoSb_3$[38,39,42–47]. To validate this concept, first-principles calculations were performed in this work. Figure 1a−c show the calculated band structures of $CoSb_3$ ($Co_4Sb_{12}$), $Fe_3CoSb_{12}$, and $CeFe_3CoSb_{12}$. It can be observed that pristine $CoSb_3$ has a low band gap of 0.135 eV, showing a typical semiconducting behavior. By substituting Fe for Co, the Fermi energy level shifts into the valence band, demonstrating its p-type semiconducting properties, albeit with a wider band gap. Further doping with Ce can narrow the band gap and allow for Fermi-level position readjustment. Thus, by adjusting the filling amount of Ce, the comprehensive thermoelectric performance of the p-type $Fe_3CoSb_{12}$ skutterudite system can be effectively regulated. It is worth noting that we also calculated the electronic structure of $CeCo_4Sb_{12}$ to validate the effect of Ce, as displayed in Supplementary Fig. 1. Ce filling can shift the Fermi level towards the conduction band, thereby demonstrating its ability to adjust the Fermi-level position to optimize the thermoelectric performance of $Fe_3CoSb_{12}$.

Moreover, unlike the use of common rigid substrates (such as silicon wafers[28], glass[32], and quartz[35], seen in Supplementary Table 1), all thin film preparation processes need to be re-explored and optimized to address the compatibility issue between skutterudite thin films and high-temperature flexible organic substrates. In this study, we utilized self-designed high-purity targets and employed advanced pulsed laser deposition (PLD) techniques to prepare p-type $Fe_3CoSb_{12}$-based flexible thermoelectric thin films (Supplementary Fig. 2), as detailed in the experimental section of the Supporting Information. Combining p-type skutterudite thermoelectric thin films with high-temperature flexible polyimide (PI) substrates. The essential requirements for the substrate include superior flexibility, insulation, and high-temperature resilience. Polyimide films aptly fulfill these requirements, hence our selection. Detailed information about polyimide film can be seen in Supplementary Fig. 3 and Supplementary Table 2, we fabricated flexible thermoelectric thin films capable of normal operation at high temperatures ($T$ > 600 K) and exhibiting high-temperature sensing functionality. We compared the maximum operating temperatures of flexible films, including organic, composite, and inorganic films, as shown in Fig. 1d. Our p-type skutterudite thermoelectric thin films have a maximum operating temperature of 653 K, which is competitively high compared to the reported operating temperatures of other films[27,48–57]. Furthermore, our films possess high flexibility and stability, with a low normalized resistance $R/R_0$ (the starting resistance is represented by $R_0$, and the resistance after 2000 repeated bending is denoted by $R$. The bending radius $r$ is 4 mm.), reaching ~1.06. Figure 1e compares the flexibility of our films with previously reported films[13,14,52–54,56,58–67], demonstrating the excellent flexibility of our p-type skutterudite films. Detailed flexibility data are provided in Supplementary Table 3[13,14,52–54,56,58–67]. More importantly, our p-type flexible skutterudite film-based device exhibits good output power performance over a broad temperature window, operating from room temperature to even over 600 K (Fig. 1f). The as-assembled device also shows a $\omega$ of 135.7 μW cm$^{-2}$ at a $\Delta T$ of 100 K with the hot-end

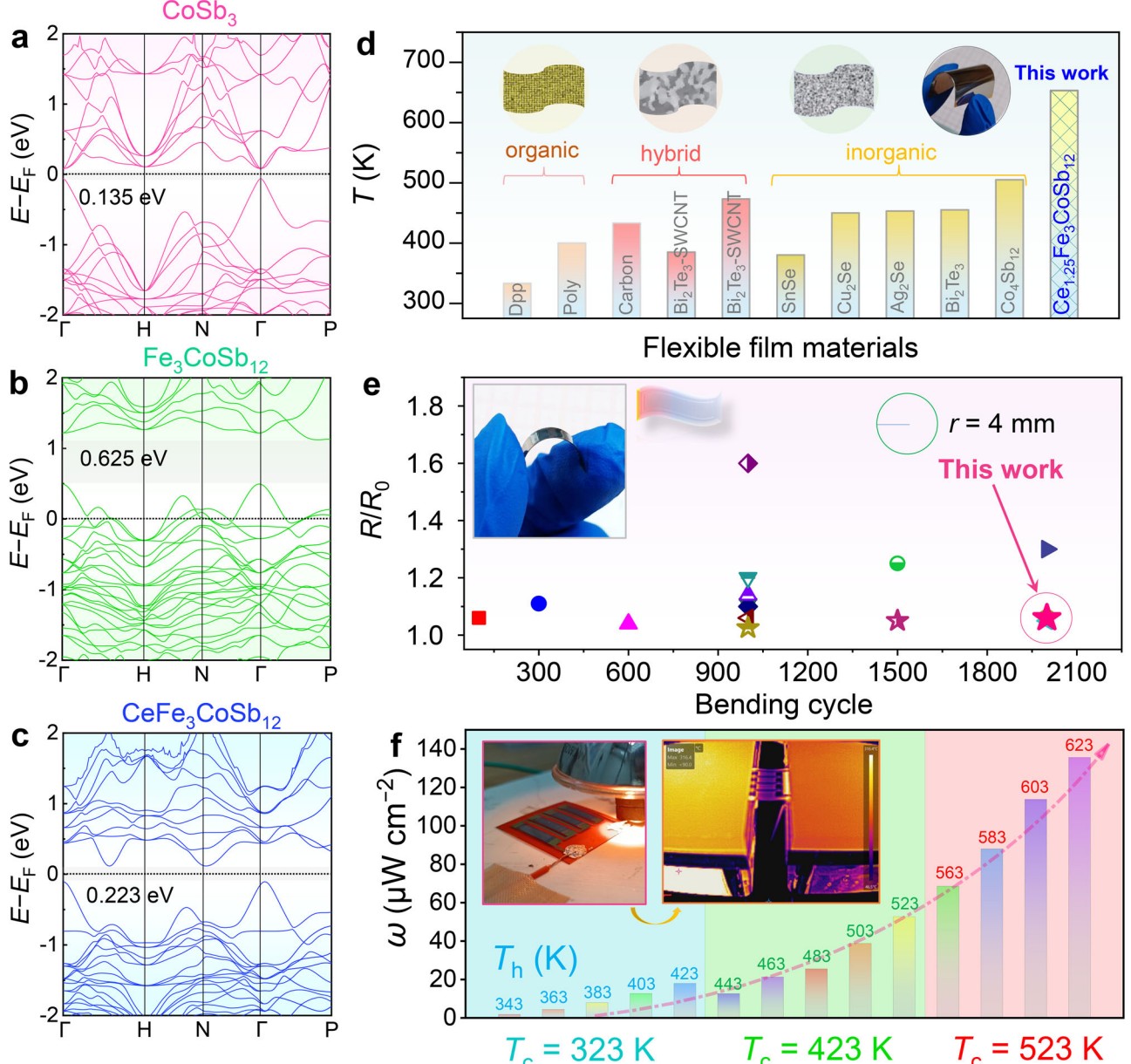

**Fig. 1 | Introduction of p-type CoSb₃-based skutterudite flexible thin films.** Calculation of band structures (**a**) CoSb₃ (Co₄Sb₁₂), (**b**) Fe₃CoSb₁₂, and (**c**) CeFe₃CoSb₁₂. (**d**) Maximum operating temperatures of various flexible thermoelectric thin films[27,48–57]. The inset schematic diagrams illustrate different types of films, along with a photograph of the p-type Ce-filled Fe₃CoSb₁₂ flexible thin film reported in this work. **e** Comparison of the normalized resistance $R/R_0$ (the starting resistance is represented by $R_0$, the resistance after 2000 repeated bending is denoted by $R$.) of the films prepared in this work with those reported[13,14,52–54,56,58–67]. The minimum bending radius $r$ in this work is 4 mm. The inset photograph illustrates the film reported in this work during bending. **f** Measured power density $\omega$ as a function of temperature difference $\Delta T$ at different hot-side temperature $T_h$ and cold-side temperature $T_c$ values between the p-type Ce-filled Fe₃CoSb₁₂ flexible thin-film-based device. The inset image presents the photographs of the thin film flexible generator and testing apparatus.

temperature $T_h$ at 623 K, showing their application prospect for power generation in the medium-to-high temperature range.

**Phase and structural characterizations**

To optimize the thermoelectric properties of Fe₃CoSb₁₂-based flexible thin films, we chose Ce as filling atoms and adjusted the doping concentration of Ce. Experimentally, we deposited five different compositions of Fe₃CoSb₁₂-based thin films on high-temperature PI substrates, defined with a nominal composition of Ce$_x$Fe₃CoSb₁₂ ($x$ = 0.25, 0.50, 0.75, 1.25, and 1.50). To investigate the phase composition and structure of the as-prepared p-type films, we first characterized all film samples using X-ray diffraction (XRD). Figure 2a shows their XRD patterns. The $2\theta$ range is from 20 ° to 67 °. By

comparing with the standard peaks of CoSb₃ (PDF # 47-1769), it can be observed that the main phase of all film samples is CoSb₃. Particularly, all samples exhibit distinct characteristic peaks such as (310), (321), and (420), demonstrating our successful preparation of CoSb₃ on flexible substrates, which is a significant breakthrough in the field. As the nominal doping content of Ce increases, the peaks of the CoSb₃ phase become more prominent, indicating an increase in the proportion of the CoSb₃ phase in the films. This confirms that the introduction of Ce can stabilize the formation of the CoSb₃ phase. Additionally, through careful comparison, we also found slight traces of impurity phases in the films, such as elemental Sb and FeSb₂ phases. This is a common occurrence during the film deposition process and is challenging to avoid. We will discuss the potential

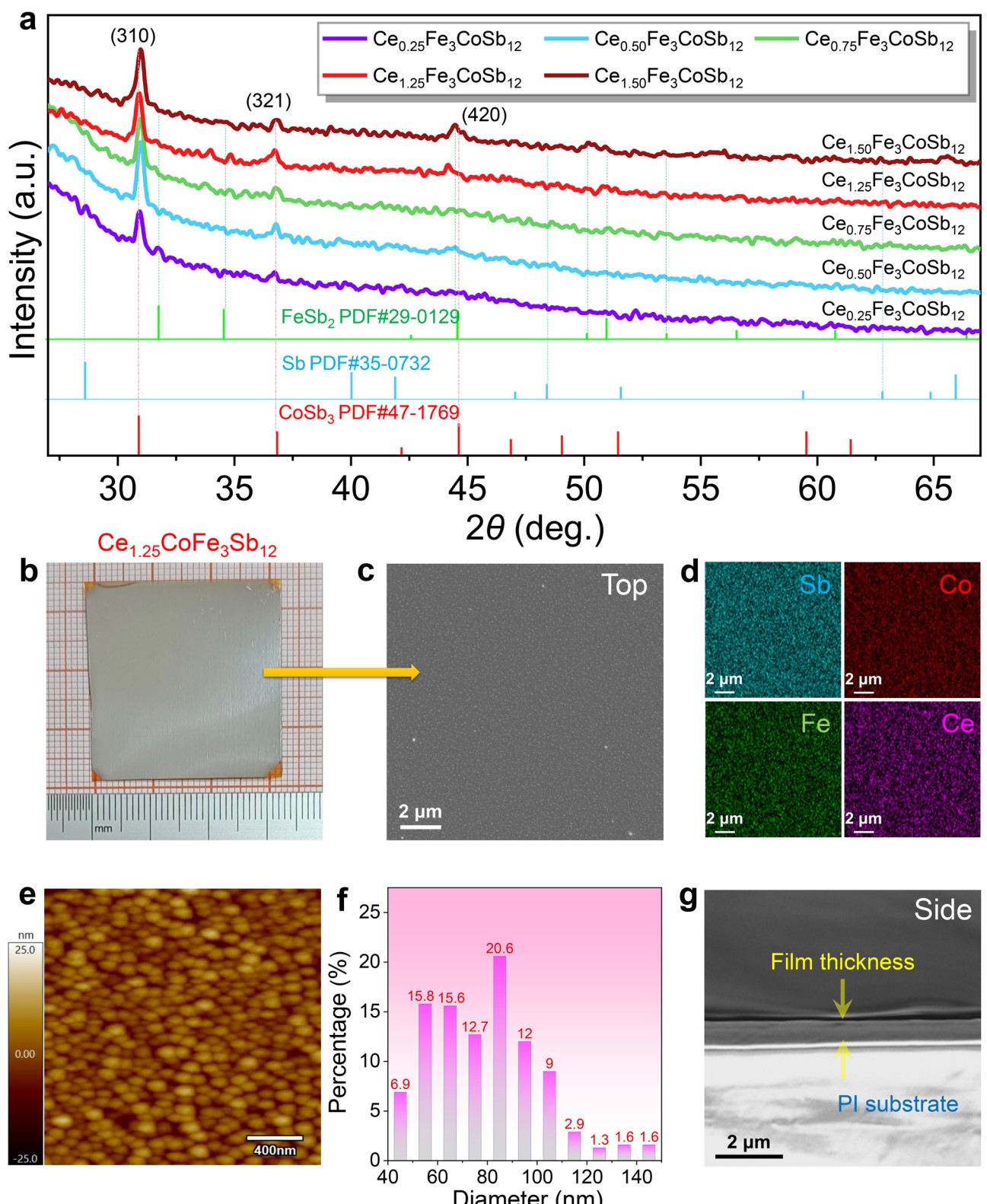

**Fig. 2 | Phase and structural characterizations of p-type CoSb₃-based skutterudite flexible thin films. a** X-ray diffraction (XRD) patterns of $Ce_xFe_3CoSb_{12}$ thin films ($x = 0.25$, 0.50, 0.75, 1.25, and 1.50) on flexible polyimide (PI) substrates. The $2\theta$ range is from 20 ° to 67 °. **b** Photograph of a $Ce_{1.25}Fe_3CoSb_{12}$ flexible thin film on a PI substrate from a top view. The size of the thin film is >30 × 30 mm². **c** Top-view scanning electron microscopy (SEM) image of $Ce_{1.25}Fe_3CoSb_{12}$ thin film and (**d**) corresponding energy dispersive spectrometry (EDS) maps for Sb, Co, Fe, and Ce. **e** Atomic force microscopy (AFM) image of $Ce_{1.25}Fe_3CoSb_{12}$ thin film. **f** Percentage distribution of nanoparticle diameters in the film structure. **g** SEM image of a $Ce_{1.25}Fe_3CoSb_{12}$ flexible thin film on a PI substrate from a cross-sectional view.

impact of these impurity phases on the performance of thermoelectric films carefully later.

To comprehensively characterize the morphology of the as-prepared flexible films, we conducted a series of micro/nanostructure characterizations on the film samples. Taking the $Ce_{1.25}Fe_3CoSb_{12}$ film as an example, Fig. 2b displays its optical image, exhibiting a typical metallic luster. The optical images of the PI substrate and the p-type film on the substrate are shown in Supplementary Fig. 4 for reference. Figure 2c presents the scanning electron microscope (SEM) top-view image of the $Ce_{1.25}Fe_3CoSb_{12}$ film, revealing a relatively smooth surface with uniformly distributed small particles, which is a typical characteristic compared to films deposited by other PLD techniques[68,69]. Supplementary Fig. 5 provides an enlarged SEM image for clearer observation of surface morphology features. Figure 2d presents the corresponding energy-dispersive X-ray spectroscopy (EDS) map of Ce, Sb, Fe, and Co. All elements are distributed uniformly at the micrometer scale, confirming the successful Ce-doping and relatively uniform composition of the as-prepared film without apparent element segregation. SEM and EDS data for $Ce_xFe_3CoSb_{12}$ films with other Ce filling contents ($x$ = 0.25, 0.50, 0.75, and 1.50) can be referenced in Supplementary Figs. 6–9. Figure 2e presents the atomic force microscope (AFM) image of the $Ce_{1.25}Fe_3CoSb_{12}$ film, revealing nanosized particle islands interconnected by atomic stacking, forming a unique nanoparticle structure, where over 80 % of the sizes are below 100 nanometers, as shown in the statistical data in Fig. 2f, sourced from AFM machine testing data. AFM data for $Ce_xFe_3CoSb_{12}$ films with other Ce doping levels can be referenced in Supplementary Fig. 10. Figure 2g shows the side-view SEM image of the $Ce_{1.25}Fe_3CoSb_{12}$ film, indicating a uniformly thick film with a thickness of approximately 300 nm. Side-view SEM images for all other films are offered in Supplementary Fig. 11 as a reference. All this evidence confirms the successful growth of $Ce_xFe_3CoSb_{12}$ films on flexible PI substrates.

## Compositional characterizations

Although the evidence from SEM-EDS indicates uniformity of all elements at the microscale, considering the typical granular morphology of the film, further investigation into the nanoscale elemental distribution is necessary. For this purpose, we employed transmission electron microscopy (TEM)-EDS to characterize the composition of localized regions of the film in detail. The TEM samples were prepared using a combination of manual peeling and alcohol sonication techniques. Figure 3a presents the TEM high-angle annular dark-field (HAADF) image of a flexible film with a nominal composition of $Ce_{1.25}Fe_3CoSb_{12}$, and the corresponding EDS maps of each element are shown in Fig. 3b (the overlap map of all elements can be seen in Supplementary Fig. 12). Here, Ce and Sb are predominantly distributed within the nanoparticles, while Co and Fe are more concentrated at the boundaries of the nanoparticles, displaying significant compositional fluctuations. However, it is important to note that these compositional fluctuations may also arise from variations in surface morphology due to non-planar surfaces. Figure 3c provides a statistical analysis of the actual compositions of Ce, Co, Fe, and Sb at 14 points in Fig. 3a. More detailed composition information is available in Supplementary Figs. 13–27 and Supplementary Table 4. The atomic percentage of the dopant element Ce fluctuates between 0.88 % and 1.98 %, remaining relatively stable overall, with an average atomic percentage exceeding 1 %, indicating successful doping. The atomic percentages of Fe and Co exhibit opposite trends, with their sum remaining essentially constant, consistent with the design principles of the nominal composition. Additionally, a comparison of the atomic percentages of Ce and Fe at different locations is presented in a line graph in Fig. 3d, illustrating the fluctuations caused by compositional variations. These rational fluctuations in composition within the nanoscale regions do not hinder the formation of the skutterudite phase, but aid in scattering phonons,

thereby potentially reducing $\kappa$. Figure 3e shows the nominal compositions of different elements in $Ce_xFe_3CoSb_{12}$ thin films ($x$ = 0.25, 0.50, 0.75, 1.25, and 1.50), while Fig. 3f presents the actual compositions of different films as determined by EDS. More detailed composition information is available in Supplementary Table 5. Significant differences between nominal and actual compositions are observed, particularly in the substitution ratio of Fe atoms for Co atoms compared to the nominal composition. However, the Fe content is sufficient to modulate pristine $Co_4Sb_{12}$ films into p-type semiconductors, achieving the desired composition. The addition of Ce is aimed at providing sufficient electrons to neutralize excess holes in the $Fe_3CoSb_{12}$ system, thereby stabilizing the formation of the p-type skutterudite phase. Hence, it is understandable for the actual doping level of Ce to be lower than its nominal concentration when the Fe content in the actual composition is lower than the nominal composition. Figure 3g compares the actual and nominal doping levels of Ce in different films, providing a more intuitive representation of their variations and differences. We predict that the actual content of Ce will play a decisive part in the thermoelectric properties of the flexible skutterudite film. Besides, we also studied the elemental composition distribution in different regions of the $Ce_{1.25}Fe_3CoSb_{12}$ film (Supplementary Figs. 28–30) and found that the distribution characteristics of elements in different regions were consistent, demonstrating the macroscopic homogeneity of the film.

## Nanostructural characterizations

In addition to compositional information, we carefully studied the nanostructure of the as-deposited films to fully understand their structural characteristics. Figure 4a presents a typical low-magnification TEM image of the $Ce_{1.25}Fe_3CoSb_{12}$ flexible film. As can be seen, the film is composed of numerous typical island-like large grains with sizes around 60 nanometers and small grains of only a few nanometers between these large grains. Figure 4b displays the corresponding selected area electron diffraction (SAED) pattern, indicating typical nano-polycrystalline features. Figure 4c displays a high-resolution TEM (HRTEM) image of one of the island-like large grains, revealing single-crystalline characteristics without apparent internal interfaces within the grain. Also, the Moiré fringes at the interfaces between this grain and others, further demonstrate the single-crystalline nature of these grains. Figure 4d depicts the HRTEM image of the small grains taken from Fig. 4a, showing clear interfaces between these small grains with different orientations, indicating the polycrystalline nanocrystalline characteristics. Figure 4e shows an HRTEM image of the small grains after tilting the zone axis, displaying the film potentially tilted to the [001] direction, which is verified via the initially indexed results. Figure 4f presents a corresponding filtered HRTEM image, confirming the potential presence of many lattice defects, such as lattice distortions and edge-like dislocations. These lattice defects combined with dense grain boundaries contribute positively to the reduction of $\kappa$ in the film. Regarding the inclusion phase, Fig. 4g displays an HRTEM image of another large grain, and the corresponding fast Fourier transform (FFT) pattern is shown as an inset image, suggesting overlapped FFT patterns. Unlike Fig. 4c, different nanoregions with different orientations coexist within this grain. Figure 4h shows a magnified HRTEM image taken from Fig. 4g, revealing relatively evenly distributed dense nanoinclusions with similar sizes of approximately 5 nm, close to quantum size. Such quantum-sized inclusions potentially contribute to the reduction of $\kappa$ in the film[9]. Figure 4i displays the corresponding filtered HRTEM image to highlight significant differences in lattice between the quantum-sized inclusions and matrix. Although we cannot directly prove the composition of these inclusion phases at such a small-scale area, all the above evidence is sufficient to illustrate the specific nanostructure features of the as-fabricated flexible films.

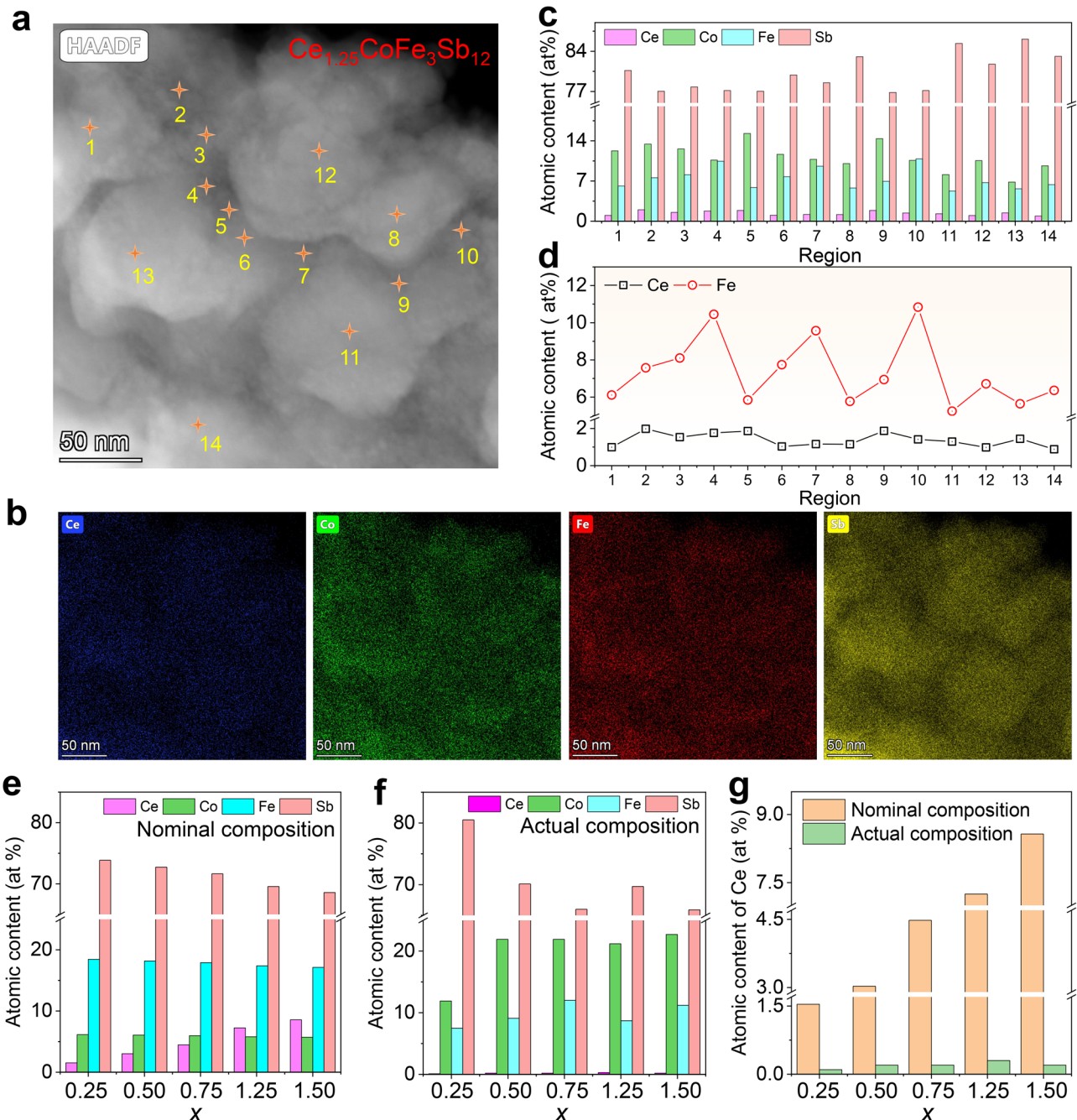

**Fig. 3 | Compositional characterizations of p-type Ce$_{1.25}$Fe$_3$CoSb$_{12}$ flexible thin film. a** Transmission electron microscopy (TEM) high-angle annular dark-field (HAADF) image of the Ce$_{1.25}$Fe$_3$CoSb$_{12}$ thin film used for compositional analysis, featuring 14 EDS points, and (**b**) corresponding EDS maps of Ce, Co, Fe, and Sb. **c** Atomic contents of different elements (Ce, Co, Fe, and Sb) obtained within different EDS spot regions in (**a**). **d** Variations of atomic contents of Ce and Fe elements obtained within different EDS spot regions in (**a**). **e** Nominal and (**f**) actual atomic contents of Ce$_x$Fe$_3$CoSb$_{12}$ thin films ($x$ = 0.25, 0.50, 0.75, 1.25, and 1.50) for Ce, Co, Fe, and Sb. **g** Comparison of nominal and actual atomic contents of Ce$_x$Fe$_3$CoSb$_{12}$ thin films ($x$ = 0.25, 0.50, 0.75, 1.25, and 1.50) for Ce.

## Thermoelectric performance

We conducted a thorough study on the thermoelectric properties of flexible films. Figure 5a–c illustrate the temperature-dependent $\sigma$, $S$, and $S^2\sigma$ of Ce$_x$Fe$_3$CoSb$_{12}$ ($x$ = 0.25, 0.50, 0.75, 1.25, and 1.50) thin films. As we can see, the $\sigma$ of all p-type films initially rises with temperature increasing, showing typical semiconductor behavior; moreover, the positive $S$ value indicates that the films are all p-type films. Additionally, $|S|$ decreases at high temperatures due to bipolar diffusion effects[70–72]. Finally, we achieve a high $S^2\sigma$ of >100 μW m$^{-1}$ K$^{-2}$ at 653 K when $x$ = 1.25. The variations in the electrical transport properties of the flexible thin-film samples are not significantly related to the changes in nominal composition. Conversely, the changes in thermoelectric properties are strongly linked to the variations in the actual composition of the flexible films (see Fig. 3f, g and Supplementary Table 5). Specifically, when the actual doping level of Ce is higher, the film exhibits significantly higher $\sigma$ and $S^2\sigma$. Additionally, due to the presence of some nanoinclusion phases within the film, such as elemental Sb and FeSb$_2$ compound (unit cells can be referred to Supplementary Figs. 31, 32), as shown by XRD and TEM results, the impact of these inclusion phases on the overall thermoelectric properties of the thin films cannot be ignored, which may lead to the considerable complexity of the electrical transport properties. We conducted first-principles calculations to determine the

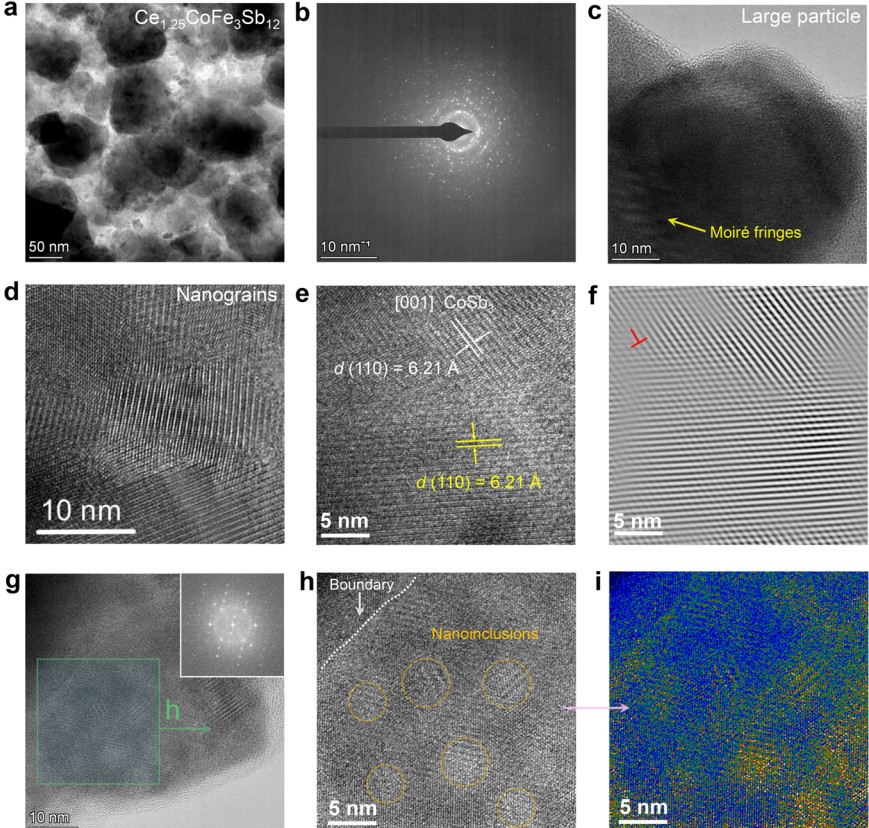

**Fig. 4 | Nanostructural characterizations of Ce$_{1.25}$Fe$_3$CoSb$_{12}$ flexible thin film.**
**a** Low-magnification TEM image of the Ce$_{1.25}$Fe$_3$CoSb$_{12}$ thin film used for nanostructure analysis. **b** Corresponding selected area electron diffraction (SAED) pattern. **c** TEM image of a large particle in the thin film structure. **d** High-resolution TEM (HRTEM) image of nanograins between large particles. **e** HRTEM image of nanograins taken from another area with indexed lattice information.

**f** Corresponding filtered HRTEM image to show lattice distortions and a potential edge-like dislocation. **g** TEM image of a large particle with quantum-dot-sized nanoinclusions in it. The inset shows a corresponding fast Fourier transform (FFT) pattern. **h** HRTEM image of these quantum-dot-sized nanoinclusions magnified from (**g**). **i** Corresponding filtered image.

electronic structures of elemental Sb and FeSb$_2$ compounds (Supplementary Figs. 33, 34). The results show that elemental Sb exhibits typical semi-metallic characteristics with overlapping conduction and valence bands, which may release free electrons to neutralize the hole carrier concentration, while the FeSb$_2$ compound is a typical semiconductor, hence its mechanism of affecting the overall performance is more complex. To understand the changes in the electrical transport properties of the thin films, Fig. 5d displays the room-temperature carrier concentration $n$ and mobility $\mu$ as a function of $x$. Generally, higher Ce content in the actual composition leads to lower $n$, confirming that the addition of Ce can indeed optimize the hole carrier concentration, reflecting the rationality of our initial design concept. Moreover, the $\mu$ of all film samples is considerably low due to the presence of numerous defects within the film, as mentioned earlier, which may scatter carriers during their transportation, leading to a significant decrease in $\mu$ and in turn $\sigma$. Especially when $x$ is 1.50, the insufficient doping of Ce in the actual composition combined with the excessive doping of Ce in the nominal composition increases the structural complexity, leading to lower $\mu$. The room-temperature effective mass $m^*$ is shown in Fig. 5e, as a function of $x$ calculated through a single parabolic band (SPB) model. Essentially, as the nominal Ce doping level increases, the $m^*$ gradually decreases, indicating the annihilation of free electrons released by the Ce doping of hole carriers. Figure 5f displays the estimated temperature-dependent $ZT$ of Ce$_{1.25}$Fe$_3$CoSb$_{12}$ flexible thin film based on its measured room-temperature $\kappa$. The inset shows the sample for $\kappa$ measurement. We used commercial equipment to test the thermal diffusivity $D$ of the film (equipment photos and testing principles can be found in Supplementary Fig. 35) and roughly calculated the $\kappa$ of the film

using the measured density and specific heat capacity $C_p$ of the target materials, with a room temperature $\kappa$ of only 0.113 W m$^{-1}$ K$^{-1}$. Due to the presence of numerous lattice defects within the film, as mentioned earlier, which may scatter phonons effectively of almost every wavelength, the low $\kappa$ is understandable. The presence of grains within 100 nanometers in flexible skutterudite films is more than 80%, effectively scattering phonons. Considering that the mean free path of phonons in CoSb$_3$ is reported around 80 nm[73,74], this scattering mechanism is pivotal in reducing thermal conductivity. It should be noted that testing temperature-dependent $D$ is challenging for flexible thin-film materials, so we only used the room temperature $\kappa$ to estimate $ZT$ at variable temperatures, obtaining a value as high as 0.6 at 653 K. Since there are very few reports about the thermoelectric properties of flexible p-type CoSb$_3$ thin films, the data we obtained is still meaningful. When determining the $\kappa$, we use the density of target materials rather than that of the thin films due to the difficulty in directly measuring the thin film density. Considering the actual granular structure of the thin film, it should result in a lower actual thin film density than the target density, therefore the actual $ZT$ of the thin film is likely to be even higher.

**Flexibility and high-temperature sensing**
Figure 6a displays photos taken before and during the bending process of Ce$_{1.25}$Fe$_3$CoSb$_{12}$ film using our self-designed automatic repeat bending film instrument and Owon BT35+ multimeter (Supplementary Movies 1, 2), demonstrating the excellent flexibility of the film. Figure 6b shows the relationship between $R/R_0$ and $r$ of the Ce$_{1.25}$Fe$_3$CoSb$_{12}$ film after 1000 bending cycles. Generally, higher $R/R_0$ usually results in a higher degree of film bending[14]. However, the

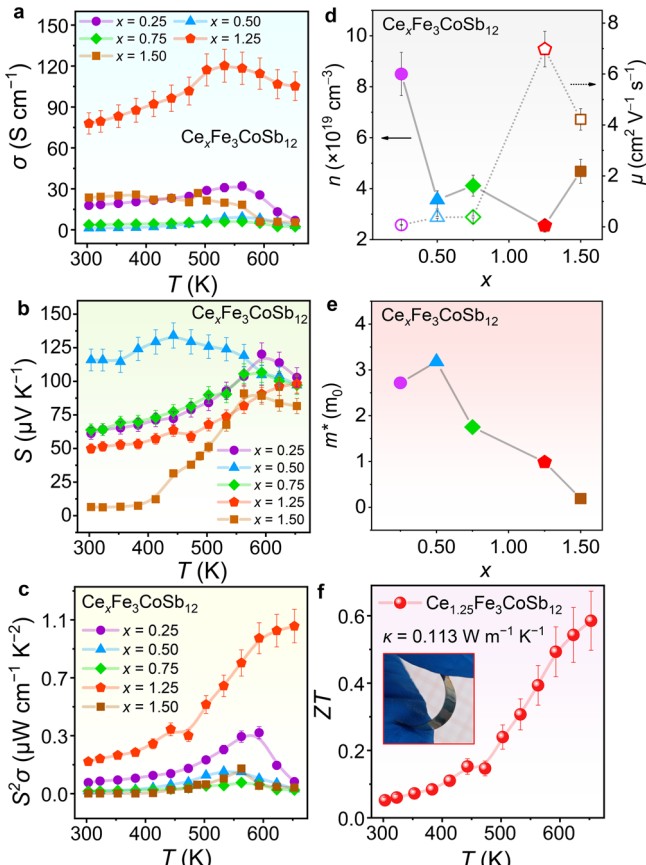

**Fig. 5 | Thermoelectric performance of p-type CoSb₃-based skutterudite flexible thin films.** Temperature-dependent (**a**) electrical conductivity $\sigma$, (**b**) Seebeck coefficient $S$, and (**c**) power factor $S^2\sigma$ of $Ce_xFe_3CoSb_{12}$ flexible thin films ($x = 0.25$, 0.50, 0.75, 1.25, and 1.50). **d** Room-temperature carrier concentration $n$ and mobility $\mu$ as a function of $x$. **e** Room-temperature effective mass $m^*$ as a function of $x$ calculated through a single parabolic band (SPB) model. **f** Estimated temperature-dependent $ZT$ of $Ce_{1.25}Fe_3CoSb_{12}$ flexible thin film based on its measured room-temperature thermal conductivity $\kappa$. The inset image displays the film sample for the $\kappa$ test.

achieved $R/R_0$ is as low as 1.05 even when $r$ equals 4 mm, proving the high flexibility and stability. Figure 6c depicts the $R/R_0$ of the $Ce_{1.25}Fe_3CoSb_{12}$ film as a function of the bending cycles, with $r$ maintained at 4 mm. It can be observed that even after 2000 repeated bending, a low $R/R_0$ of approximately 1.06 can still be achieved, demonstrating extraordinary flexibility and durability. Additionally, for reference, Supplementary Fig. 36 displays photos of film bending under different $r$ values. The excellent flexibility of the films can be attributed to their structure: the films have uniformly dense nano-sized grains with good crystallinity. Through the optimization of deposition, the film thickness was controlled to be thin, approximately 300 nm thick, while the selected flexible substrate is only 125 μm thick. The strong bond between the film and the flexible substrate also enhances flexibility. Figure 6d displays a photo of the self-built light-thermal detection platform. By adjusting the power of the DC power supply, the temperature of the lamp can be controlled. Based on the Seebeck effect, the light spot converges on one side of the flexible thermoelectric film to generate detection signals. More detailed information about the detection platform is shown in Supplementary Fig. 37 for reference. Figure 6e shows the amplified image of the light-thermal detection process. Additionally, an infrared thermal imager captures the temperature distribution on the flexible thermoelectric film during the light-thermal detection

process, as shown in Fig. 6f. It can be observed that there is a significant temperature difference distribution on both sides of the flexible film. The current signal variation over time during the detection process can be obtained by turning the lamp on and off. Figure 6g shows the relationship between the measured current $I$ and the time the focused light beam strikes the flexible thermoelectric film. When the hot side of the thermoelectric film is irradiated by thermal light, current is generated through the thermoelectric film, while the other side remains at ambient room temperature (25 °C). The response speed is considerably fast. As the temperature increases, the detected $I$ gradually increases. When the center temperature of the light spot is 473 K, 523 K, and 643 K, the peak $I$ is 1.8 μA, 2 μA, and 3 μA, respectively. Furthermore, the variation of the $I$ during the detection process under different hot-side temperatures ranging from ambient room temperature (25 °C) to medium-high temperatures is shown in Supplementary Fig. 38 for reference (Supplementary Movie 3 demonstrates detailed information about the platform components, while Supplementary Movies 4, 5 show measurement results). Figure 6h shows the variation of the detected $I$ and voltage $V$ signals with temperature from 300 K to 675 K. This significantly expands the detection temperature range of flexible thermoelectric films and is of great significance for future practical applications, serving as high-temperature flexible sensors. The excellent detection performance is attributed to the good thermoelectric properties of the flexible thermoelectric film as discussed above. The flexible skutterudite thermoelectric film enables thermoelectric detection across a broad temperature spectrum, from room temperature to medium-to-high levels. Its outstanding features, including high-temperature resistance, flexibility, and lightweight nature, make it suitable for applications in environments with complex shapes, particularly in medium and high-temperature settings. Fire-resistant clothing plays a crucial role in safeguarding firefighters during firefighting endeavors. Leveraging the flexible thermoelectric film, a self-powered intelligent fire alarm system can be developed. This system can preemptively alert firefighters before their attire is compromised by fire, allowing them to take timely preventive measures to avoid burns[75,76]. Utilizing thermoelectric film to enable precise temperature sensing and responsive fire alarm capabilities in combustible materials holds significant importance for establishing a safe home environment[77,78]. Moreover, a basic thermoelectric hydrogen sensor can be created by applying a catalyst to one side of the thermoelectric film. When exposed to a hydrogen-containing environment, the catalyst facilitates the reaction between hydrogen and oxygen, producing water vapor and releasing heat. Consequently, the end with the catalytic metal deposit becomes hot, while the end without it remains cooler. This temperature gradient generates an electrical signal through the Seebeck effect, enabling the detection of hydrogen concentration.

## Fabrication and performance evaluation of generator

To validate the practical application potential of the prepared flexible thermoelectric thin film, we, for the first time, fabricated four-leg $Ce_{1.25}Fe_3CoSb_{12}$ thin-film-based flexible thermoelectric devices and systematically discussed their output performance. The construction process of the thin-film-based device is illustrated in Supplementary Fig. 39 for reference. Figure 7a displays the structure of the as-designed device, where four $Ce_{1.25}Fe_3CoSb_{12}$ thin films are combined via Au electrodes and wires using the traditional four-wire method on a flexible PI substrate. Figure 7b displays the self-built testing platform used to measure the output performance of our designed flexible device (Supplementary Fig. 40 provides detailed information on each part of the platform, and Supplementary Movie 6 demonstrates the measurement process of the device). Figure 7c shows the temperature distribution on the device captured by an infrared camera during the measurement process. It can be observed that there is a significant

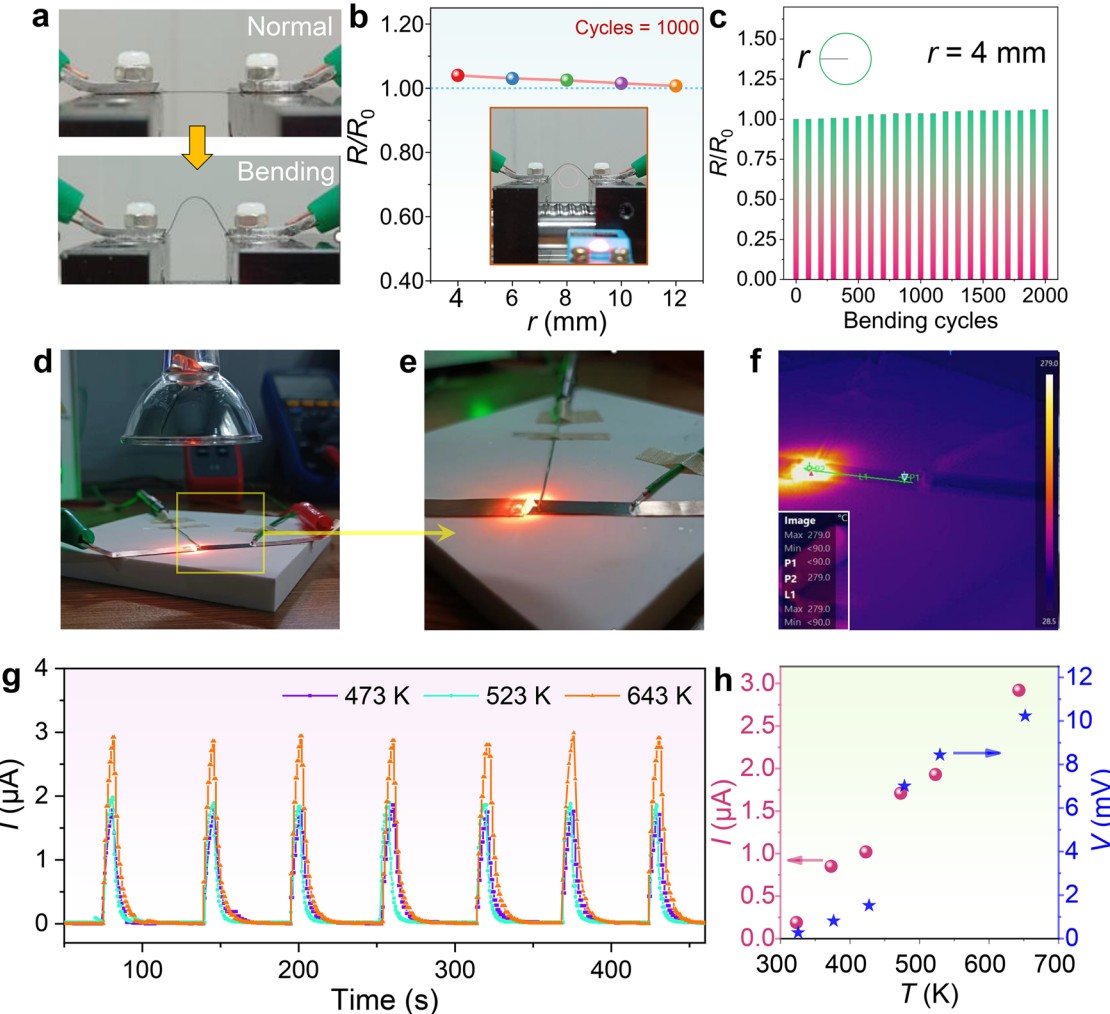

**Fig. 6 | Flexibility and high-temperature sensing potential of p-type Ce$_{1.25}$Fe$_3$CoSb$_{12}$ flexible thin film. a** Photographs of the Ce$_{1.25}$Fe$_3$CoSb$_{12}$ flexible thin film before and during bending. **b** The relationship between $R/R_0$ and $r$ after 1000 bending cycles. **c** $R/R_0$ as a function of bending cycles. Here $r$ is 4 mm. **d** Photograph of the platform used to evaluate the sensing characteristics of the thin film. **e** Photograph of the setup for illumination testing. The light beam focuses on the side of the thin film and generates a temperature gradient. **f** Temperature distribution on the thin film under infrared imaging. **g** Measured $I$ as a function of time when low frequency is introduced for optical pulse modulation. **h** $I$ and voltage $V$ values at different light temperatures.

temperature difference distribution on both the hot and cold sides of the flexible film device. Change in temperature differences ($\Delta T$) can be obtained by keeping one side of the heating plate at a specified temperature and the other side at another specified temperature, with the hot-side temperature $T_h$ ranging from 563 to 623 K, and the cold-side temperature $T_c$ maintained at 523 K. Supplementary Fig. 41 displays additional temperature difference distributions. Figure 7d shows the experimentally measured open-circuit voltage $V_{oc}$ as a function of $\Delta T$. As $\Delta T$ increases from 40 K to 100 K, the $V_{oc}$ increases from 26.7 mV to 37.5 mV. Figure 7e, f show the relationships between the $V_{oc}$ and the output power $P$ as functions of the load current $I_{load}$ measured at different $\Delta T$s, where the $P$ values are 12.3, 15. 8, 20.4, and 24.4 nW at $\Delta T$s of 40, 60, 80, and 100 K, respectively. Additionally, it is noteworthy that power density $\omega$ is one of the most valuable indexes for evaluating the performance of thermoelectric devices. Figure 7g illustrates the relationship between $\omega$ and $\Delta T$. The $\omega$ monotonically increases with increasing the $\Delta T$, reaching 68.6, 88.0, 113.7, and 135.6 μW cm$^{-2}$ at $\Delta T$s of 40, 60, 80, and 100 K, respectively. Additionally, we conducted similar measurements by setting the $T_c$ to 323 and 423 K, as shown in Supplementary Figs. 42, 43. Our flexible thin-film-based device exhibits better performance at high temperatures due to the higher thermoelectric properties of the thin films, which can also

be confirmed via Supplementary Figs. 44, 45. Figure 7h illustrates the operation of the Ce$_{1.25}$Fe$_3$CoSb$_{12}$ flexible thin-film-based device in harnessing waste heat for power generation under a curved high-temperature surface scenario. The flexible thin-film device was securely affixed to the surface of the high-temperature furnace tube within the tubular furnace and consistently produced a relatively stable current close to 0.6 μA and voltage near 15 mV, as depicted in Fig. 7i. The temperature of the high-temperature side of the flexible device exceeds 600 K, as evidenced by the infrared image showing the temperature distribution inserted in Fig. 7i. For clearer insights, Supplementary Fig. 46 and Supplementary Movie 7 provide detailed views of the inset pictures. This is the first successful fabrication of p-type flexible CoSb$_3$-based film devices, achieving excellent performance over such a wide temperature range from medium to high temperatures, indicating that our designed CoSb$_3$-based flexible thermoelectric devices are particularly effective for medium-to-high-temperature scenarios, filling the gap in high-temperature flexible thermoelectric power generation.

In this study, a p-type CoSb$_3$-based skutterudite flexible thin films, nominally composed of Ce$_{1.25}$Fe$_3$CoSb$_{12}$ with an enhanced $S^2\sigma$ of >100 μW m$^{-1}$ K$^{-2}$ and an approximated $ZT$ of ~0.6 at 653 K, has been fabricated using an advanced PLD technique with a self-designed

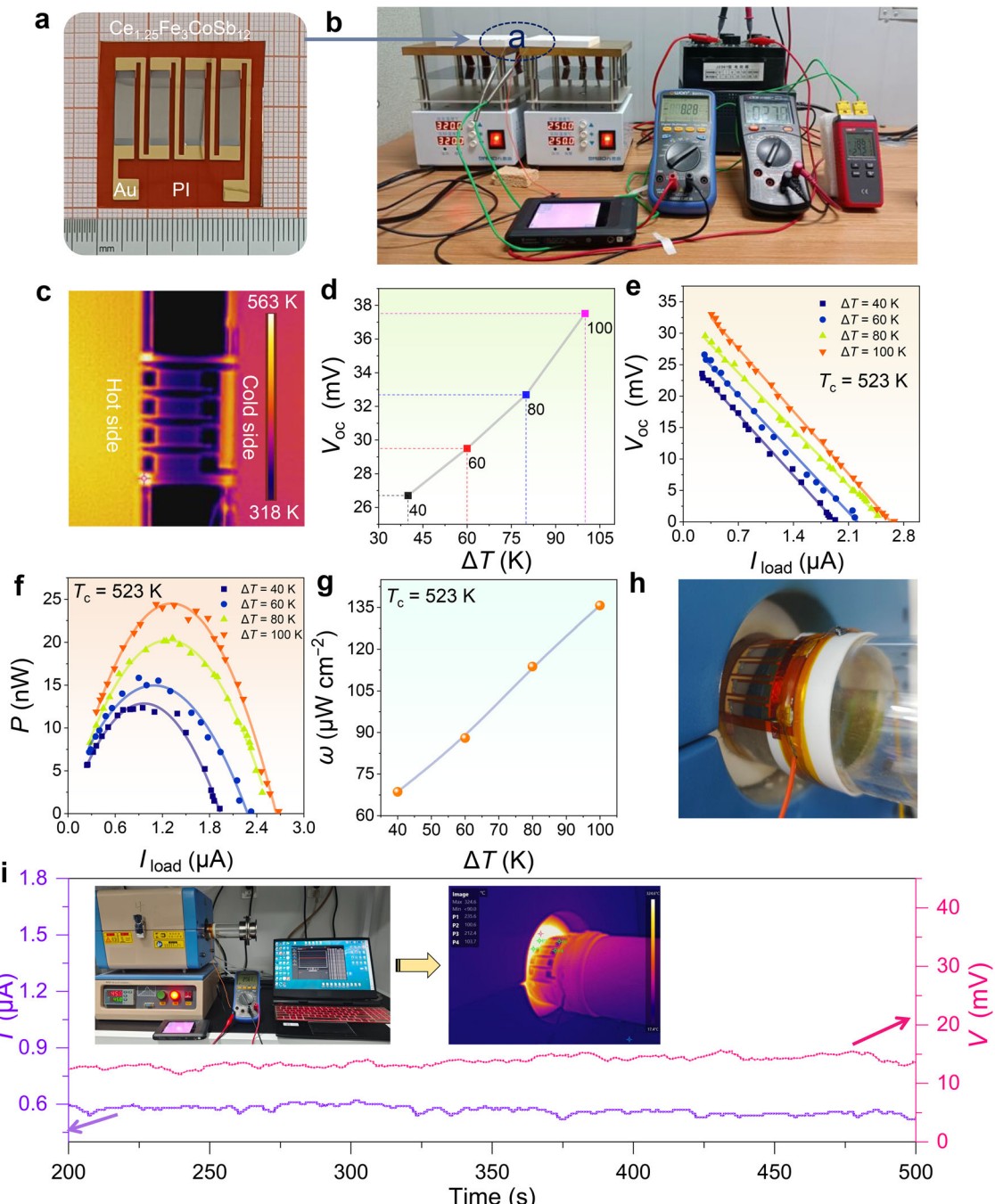

**Fig. 7 | Fabrication and performance evaluation of p-type Ce$_{1.25}$Fe$_3$CoSb$_{12}$ thin-film-based flexible thermoelectric generator. a** Photograph exhibiting the flexible Ce$_{1.25}$Fe$_3$CoSb$_{12}$ thin-film-based flexible generator. **b** Photograph exhibiting the testing platform used to assess the performance of the manufactured flexible thin film generator. **c** Infrared photograph of temperature distribution on the flexible device after applying a temperature difference $\Delta T$. **d** The open-circuit voltage $V_{oc}$ of the test varies with $\Delta T$. The cold-side temperature $T_c$ is 523 K. The (**e**) $V_{oc}$ and (**f**) output power $P$ of the test vary with load current $I_{load}$ at different $\Delta T$s. The $T_c$ is 523 K. **g** The measurement output power density $\omega$ varies with $\Delta T$. Here the $T_c$ is 523 K. **h** Image of the Ce$_{1.25}$Fe$_3$CoSb$_{12}$ flexible thin-film-based device securely adhered to the surface of a heating tube furnace. **i** Measurement of $I$ and $V$ over time as the device gathers residual heat from the curved surface of the tube furnace for power generation. An inset displays the photograph of the platform utilized for measuring $I$ and $V$ during waste heat power generation. Another inset presents an infrared image illustrating the surface temperature distribution of the flexible thin-film-based device during waste heat power generation.

target material, resulting in distinctive compacted polycrystalline nanostructures. Leveraging the compact nanostructures and thin, flexible PI substrates, we achieve a low $R/R_0$ of ~1.06 after 2000 repeated bending under a minimum $r$ of only 4 mm, showing unprecedented flexibility. Moreover, we, for the first time, fabricate a p-type Ce$_{1.25}$Fe$_3$CoSb$_{12}$ thin-film-based flexible generator, demonstrating a high $\omega$ of 135.7 µW cm$^{-2}$ at a $\Delta T$ of 100 K and a $T_c$ of 523 K. This study fills the gap of high-temperature flexible thermoelectric power generation

and provides insights for designing other high-temperature flexible thermoelectric thin films for practical applications.

## Methods
### Materials
Sb (purity 99.995%, diameter: 1–10 mm) was purchased from CNBM (Chengdu) Optoelectronic Materials Co., Ltd. Co (purity 99.95%, diameter: 3–10 mm) was purchased from Beijing Gold Crown for The New

Material Technology Co., Ltd. Fe (purity 99.95%, diameter: 3–10 mm) was purchased from Beijing Gold Crown for The New Material Technology Co., Ltd. Ce (purity 99.95%, diameter: 1–10 mm) was purchased from ZhongNuo Advanced Material (Beijing)Technology Co., Ltd. PI substrate was purchased from Shenzhen Runsea Electronic Co., Ltd.

## Fabrication of p-type $CoSb_3$-based flexible thin films

Thin films of $Ce_xFe_3CoSb_{12}$ ($x = 0.25, 0.50, 0.75, 1.25,$ and $1.50$) were deposited onto flexible PI substrates capable of withstanding temperatures up to 400 °C (673 K). The deposition process utilized a PLD system (Neocera 120) with self-made targets. For instance, a high-purity (99.98%) Ce-doped $Fe_3CoSb_{12}$ target was obtained via the temperature gradient zone melting (TGZM) route[26,70,72,79,80]. PI substrates with sizes of 33 mm × 33 mm × 0.125 mm underwent ultrasonic cleaning for 20 min in acetone, 10 min in absolute ethyl alcohol, and 10 min in deionized water, followed by drying in an oven. The PLD chamber was evacuated to a base pressure of $1.5 \times 10^{-4}$ Pa. Targets were positioned in a holder within the chamber and traversed for uniform ablation of the target surface, ensuring uniform film growth. The distance between the target and the substrate was approximately 10 cm. Substrates were heated using the built-in heater within a range of 20 ~ 800 °C. Heating occurred from room temperature to 523 K at a rate of 10 K min$^{-1}$ and maintained during deposition. The pulsed laser employed was a krypton fluoride excimer laser operating at a 10 Hz repetition rate, 20 ns pulse duration, 248 nm wavelength, and a laser power of $190 \pm 5$ mJ per pulse. Subsequently, annealing took place for 35 min in an Ar ambient. The temperature increased at a rate of 10 K min$^{-1}$, with the annealing temperature set at 250 °C, followed by natural cooling to room temperature.

## Assembly of $CoSb_3$ thin-film-based flexible thermoelectric devices

To begin, the electrode mask was placed onto a clean and dry flexible PI substrate and securely fastened using high-temperature tape. Subsequently, a layer of Au electrode, approximately 200 nm thick, was deposited within a vacuum chamber. Following this step, the mask plate was removed, and the substrate was covered with a thin-film thermoelectric arm mask plate, which was also affixed using high-temperature tape and inserted into the vacuum chamber. $CoSb_3$-based thermoelectric targets were then deposited onto the substrate using PLD. Finally, the film-electrode integrated thermoelectric device was extracted and transferred to a tube furnace for annealing under continuous argon gas protection.

## Characterizations

The morphology of the as-grown thin films was examined using field emission SEM (FESEM, Tescan Lyra-3) equipped with EDS operating at 20 kV. Crystalline phases were identified via grazing incidence XRD (GIXRD, XRD-7000S/L) at 2 ° with a $2\theta$ angle range of 20 ~ 67 ° and a scanning velocity of 3°min$^{-1}$, utilizing Cu Kα radiation ($\lambda = 0.15406$ nm). The nanostructure was investigated using a TEM (Talos F200x). The temperature distributions of thin films and devices were acquired using a high-performance compact thermal imaging camera for professionals (HM-TPK20-3AQF/W, HIKMICRO Pocket2). AFM (Dimension Icon, Bruker) was utilized to examine the surface morphology in tapping mode. The tapping mode was employed to perform nanoscale topographic characterization of the thin film surface. The tip curvature radius is 35 nm with the type of NSG03/Au.

## Performance evaluation of thin films

The $S$ and $\sigma$ of the films were concurrently measured using an MRS-3 thin-film thermoelectric test system (Wuhan Joule Yacht Science & Technology Co., Ltd., China). The Hall coefficient $R_H$ was determined using the electrical transport properties measurement system (ET9105-HS, East Changing Technologies, Inc.) at room temperature.

The signified $D$ data of thin films on a substrate were measured using an AC method thermal diffusivity measurement system (Laser PIT, ADVANCE RIKO) under vacuum conditions, with dimensions of 18.05 mm × 8.75 mm × 300 nm. The $C_p$ data were calculated using the SPB model[81], while the mass density $d$ data were obtained from the $d$ of the target materials via the Archimedes method[82].

## Sensing potential evaluation of thin films

For the heat signal response test, light and heat were provided by a halogen tungsten lamp (24 V, 150 W) utilized to drive the lamp. Current $I$ as a function of time was recorded using a Digital Multimeter with Bluetooth (Owon BT35 + ).

## Performance evaluation of thermoelectric generators

The power generation performance of the 4-leg thermoelectric device was monitored using a Digital Multimeter with Bluetooth (Owon BT35 + ) employing the 4-wire method. The $\Delta T$ was regulated using laboratory-made equipment, which controlled the $T_h$ and $T_c$, along with a variable resistor to adjust the device resistances to maximize the output power.

## Data availability

The data generated in this study are provided in the Source Data file. Source data are provided with this paper.

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

## Acknowledgements

This work was supported by the Research Fund of the National Natural Science Foundation of China (51774239), and the Research Fund of the State Key Laboratory of Solidification Processing in NWPU (2022-TS-03). Zhi-Gang Chen thanks the financial support from the Australian Research Council, HBIS-UQ Innovation Centre for Sustainable Steel project, and QUT Capacity Building Professor Program. Zhigang Chen and Meng Li acknowledge the National Computational Merit Allocation Scheme 2024 (wk98), sponsored by National Computational Infrastructure, for providing computational resources and services. This work was enabled by the use of the Central Analytical Research Facility hosted by the Institute for Future Environments at QUT.

## Author contributions

D.Li & X.-L.Shi contributed equally to this work. Z.-G.Chen, S.-M.Li and H.Zhong supervised the project and conceived the idea. D.Li and X.-L.Shi designed the experiments and wrote the manuscript. D.Li, J.-X.Zhu, Z.-K.Han, Z.-Y.Feng, Y.-X.Chen and J.-Y.Wang performed the sample synthesis, structural characterization, and thermoelectric transport property measurements. M.Li and X.-L.Shi undertook the theoretical work. J.-X.Zhu and X.Ma conducted the TEM measurements and T.-Y.Cao conducted the measurement of the in-plane $\kappa$ of the films. X.-L.Shi, M.Li, W.-D.Liu, Z.-G.Chen, and S.-M.Li undertook the thermoelectric performance evaluation. All the authors discussed the results and commented on the manuscript. All authors have approved the final version of the manuscript.

## Competing interests

The authors declare no competing interests.
