## [Peer Review File · Nature Communications]

High-performance flexible p-type Ce-filled Fe₃CoSb₁₂ skutterudite thin film for medium-to-high-temperature applicationsREVIEWER COMMENTS

Reviewer #1 (Remarks to the Author):

This paper reports the preparation of a novel p-type inorganic flexible thin film, achieving high thermoelectric performance and excellent flexibility. Previously, the application of p-type flexible inorganic thermoelectric thin films in the medium to high-temperature range has been rare, thus, this work is highly novel and may even pioneer a new research field, namely flexible inorganic thermoelectric thin films for medium to high-temperature applications. They also fabricated a high-temperature flexible device with high output power density to demonstrate the importance of this material and device in flexible thermoelectric power generation applications at medium to high temperatures. This is crucial guidance for expanding the application of flexible thin film thermoelectrics in high-temperature domains. Essentially, the experimental data in this paper are rich and consistent with theoretical calculations, and the logic is clear. The characterization section of the paper is detailed and accurate, employing numerous structural characterizations such as high-resolution electron microscopy, along with first-principles computational assistance to elucidate performance changes, thus ensuring the completeness and reliability of the data. The publication of this paper is anticipated to have a significant impact on the thermoelectric field. Therefore, I recommend minor revisions before accepting this paper..

1. When choosing polyimide film as substrate, in addition to considering flexibility, what other requirements do the substrate need to meet?
2. Both Fe and Ce are used to optimize the performance of skutterudite thin film. Why are these two elements chosen and what roles do they play?
3. The thermal conductivity is very low. This is important for a high ZT. Accurate in-plane thermal conductivity measurement of thin films is known to be very difficult. More discussions are needed for the measurement mechanism.
4. Infrared pictures are used in Figure 1, Figure 6 and Figure 7 of the manuscript. Please write clearly the model of infrared photographic equipment in the experimental section.

Reviewer #2 (Remarks to the Author):

The authors present a high-performance flexible p-type Ce-filled Fe₃CoSb₁₂ Skutterudite thin film for medium-to-high-temperature applications. I have carefully read this paper. Based on my experience, this paper appears to be the first systematic report and invention of a flexible inorganic thin film suitable for medium-to-high temperature ranges, which is very intriguing. Through a series of process optimizations, the authors have obtained a stable p-type thermoelectric thin film, with a high-temperature *ZT* value approaching 0.6, a rare achievement. The subsequent light source detection and device design indirectly demonstrate the higher thermoelectric performance of the film at high temperatures, and the application of the designed device in flexible environments at medium-to-high temperatures is also highly practical. Additionally, considering the high quality of the paper writing, ample evidence provided, vivid and illustrative images, coherent logic, comprehensive and well-supported performance analysis, and calculation-assisted proof of data patterns, as well as the appropriately summarized tables, this paper is suitable for publication in Nature Communications and is expected to attract significant attention and citations. I have a few

minor suggestions for revisions:

1. For Figure 2e, what is the imaging mode of AFM? Additionally, if discussing the size of surface particles, the numerical value of the probe's tip radius curvature should be provided to demonstrate that the tip curvature radius is small enough, thereby ensuring accurate determination of particle relative sizes.
2. Defects and nanoinclusion can significantly affect the property. To better understand the structure characteristics, it is better to briefly introduce the filtering process from Figure 4h to Figure 4i and highlight the related importance.
3. Please provide further discussions on strategies for enhancing ZT and the application potentials in the sensor.
4. There are minor grammatical errors that can be corrected.

Response to reviewers' comments

Reviewer #1

This paper reports the preparation of a novel p-type inorganic flexible thin film, achieving high thermoelectric performance and excellent flexibility. Previously, the application of p-type flexible inorganic thermoelectric thin films in the medium to high-temperature range has been rare, thus, this work is highly novel and may even pioneer a new research field, namely flexible inorganic thermoelectric thin films for medium to high-temperature applications. They also fabricated a high-temperature flexible device with high output power density to demonstrate the importance of this material and device in flexible thermoelectric power generation applications at medium to high temperatures. This is crucial guidance for expanding the application of flexible thin film thermoelectric in high-temperature domains. Essentially, the experimental data in this paper are rich and consistent with theoretical calculations, and the logic is clear. The characterization section of the paper is detailed and accurate, employing numerous structural characterizations such as high-resolution electron microscopy, along with first-principles computational assistance to elucidate performance changes, thus ensuring the completeness and reliability of the data. The publication of this paper is anticipated to have a significant impact on the thermoelectric field. Therefore, I recommend minor revisions before accepting this paper.

Author reply: We greatly appreciate the positive comments and constructive suggestions, which are invaluable for enhancing the quality of our manuscript.

Comment 1: When choosing polyimide film as substrate, in addition to considering flexibility, what other requirements do the substrate need to meet?

Author reply: We acknowledge the essential requirements for the substrate, including superior flexibility, insulation, and high-temperature resilience. The comprehensive test reports on polyimide substrates are delineated in **Table R1**. Flexibility stands as a pivotal attribute, which is crucial for conforming to diverse and intricate shapes. Moreover, the substrate must ensure insulation to preserve the integrity of the skutterudite film during electrical transport performance tests, thereby ensuring the precision of electrical measurements. High-temperature resistance (exceeding 600 K) serves as another critical criterion for substrate selection, given the intended application of the skutterudite film in high-temperature environments. Consequently, the substrate must harmonize with the thermoelectric film's high-temperature resilience. Polyimide films aptly fulfill these requirements, hence our selection. We have included this information on **Page 6** of the revised manuscript.

Table R1. (Supplementary Table 2). Detailed information of flexible polyimide substrate.

No.	Test content name		Unit	Tolerance range	Test result
1	Thickness		μm	125	125
2	Thickness tolerance range		μm	±5	—
3	Tensile strength	MD	MPa	≥135	139
		TD	MPa	≤115	118
4	Elongation	MD	%	≥35	55
		TD	%		70
5	Insulation strength	average	MV/m	≥135	135
		individual	MV/m	≥100	110
6	Surface resistivity (200°C)		Ω	≥1.0*10 ¹³	1.1*10 ¹³
7	Shrinkage (200°C ± 5)	MD	‰	≤1.0	0.3
		TD		≤0.9	0.12
8	Breakdown voltage	thickness (125μm)	KV	≥5	5
9	Temperature resistance	thickness (125μm)	°C	350	350
10	Volume resistivity (200°C)		Ω m	≥1.0*10 ¹³	1.0*10 ¹³

Comment 2: Both Fe and Ce are used to optimize the performance of skutterudite thin film. Why are these two elements chosen and what roles do they play?

Author reply: Initially, we formulate the nominal composition to produce p-type thin films based on CoSb_3 . Due to Fe possessing one fewer 3d electron compared to Co, the substitution of Fe for Co within the $\text{Co}_4\text{Sb}_{12}$ structure can introduce one hole into the valence band. Consequently, the substitution of Fe for Co facilitates the formation of p-type skutterudite¹⁻⁷. Doping with Fe leads to significant instability in the formation of the skutterudite phase, primarily due to charge imbalance^{4,5,8}. Hence, we introduce high-charge-state filler atoms to counterbalance the high hole concentration and achieve electrical neutrality conditions in the p-type skutterudite system. Ce has been identified as an excellent filler candidate for skutterudite due to its exceptional performance and cost-effectiveness in bulk materials. Therefore, the combined control of Ce-filling and Fe-substitution for Co aims to modulate the p-type thermoelectric properties of the pristine CoSb_3 . First-principles calculations were conducted in this study to validate this concept. **Figs. R1a-c** depict the calculated band structures of CoSb_3 ($\text{Co}_4\text{Sb}_{12}$), $\text{Fe}_3\text{CoSb}_{12}$, and $\text{CeFe}_3\text{CoSb}_{12}$. It is evident that pristine CoSb_3 exhibits a narrow band gap of 0.135 eV, indicative of typical semiconducting behavior. Upon substituting Fe for Co, the Fermi energy level shifts into the valence band, indicating its p-type semiconducting properties, albeit with a wider band gap. Further doping with Ce can narrow the band gap and facilitate Fermi-level adjustment. Thus, by controlling the amount of Ce filling, the overall thermoelectric performance of the p-type $\text{Fe}_3\text{CoSb}_{12}$ skutterudite system can be effectively tuned. Ce filling can shift the Fermi level towards the conduction band, demonstrating its capability to optimize the Fermi-level position for enhancing the thermoelectric performance of $\text{Fe}_3\text{CoSb}_{12}$.

Fig. R1. (Fig. 1 in the revised manuscript). The composition design is carried out by band structures calculation under DFT.

Comment 3: The thermal conductivity is very low. This is important for a high ZT . Accurate in-plane thermal conductivity measurement of thin films is known to be very difficult. More discussions are needed for the measurement mechanism.

Author reply: To accurately measure the thermal diffusivity of the thin film material, we utilized the AC Method Thermal Diffusivity Measurement System (LaserPIT) to test the in-plane thermal diffusivity of our film and then determine the thermal conductivity. By matching the electrical performance of the film in the same direction, we obtained a more precise in-plane ZT value. This system employs the AC method, known as the Angstrom method, for scanning laser heating to measure the thermal diffusivity of sheet materials in the in-plane direction. It enables accurate measurement of thermal diffusivity for a wide range of sheet materials. The principle of measurement involves determining the thermal conductivity of a thin film deposited on a specialized test substrate by measuring the thermal diffusivity of both the deposited and non-deposited areas on the same side of the substrate. The thermal conductivity of the thin film is evaluated based on the measurement results

from both areas, as well as the thickness and volume-specific capacity of the glass substrate and the thin film.

Comment 4: Infrared pictures are used in Figure 1, Figure 6 and Figure 7 of the manuscript. Please write clearly the model of infrared photographic equipment in the experimental section.

Author reply: The temperature distribution of the thin films and devices depicted in **Fig. 1**, **Fig. 6**, and **Fig. 7** of the manuscript was captured using a high-performance compact thermal imaging camera designed for professionals (HM-TPK20-3AQF/W, HIKMICRO Pocket2). Additional details regarding this are now included on **Page 19** of the revised manuscript.

Reviewer #2

The authors present a high-performance flexible p-type Ce-filled $\text{Fe}_3\text{CoSb}_{12}$ Skutterudite thin film for medium-to-high-temperature applications. I have carefully read this paper. Based on my experience, this paper appears to be the first systematic report and invention of a flexible inorganic thin film suitable for medium-to-high temperature ranges, which is very intriguing. Through a series of process optimizations, the authors have obtained a stable p-type thermoelectric thin film, with a high-temperature ZT value approaching 0.6, a rare achievement. The subsequent light source detection and device design indirectly demonstrate the higher thermoelectric performance of the film at high temperatures, and the application of the designed device in flexible environments at medium-to-high temperatures is also highly practical. Additionally, considering the high quality of the paper writing, ample evidence provided, vivid and illustrative images, coherent logic, comprehensive and well-supported performance analysis, and calculation-assisted proof of data patterns, as well as the appropriately summarized tables, this paper is suitable for publication in Nature Communications and is expected to attract significant attention and citations. I have a few minor suggestions for revisions.

Author reply: We are grateful for the positive feedback and constructive suggestions, which greatly enhance the quality of our manuscript.

Comment 1: For Figure 2e, what is the imaging mode of AFM? Additionally, if discussing the size of surface particles, the numerical value of the probe's tip radius curvature should be provided to demonstrate that the tip curvature radius is small enough, thereby ensuring accurate determination of particle relative sizes.

Author reply: The tapping mode of an atomic force microscopy (AFM) was employed to perform

nanoscale topographic characterization of the thin film surface. With a tip curvature radius of 35 nm, significantly smaller than the size of nanoparticles collected on the film surface, the test ensures accuracy and precision. Furthermore, relevant parameters for the AFM tips used in this study are provided in **Table R2** for clarity. This information has been added to **Page 19** of the revised manuscript.

Table R2. The type and parameters of the tip used in the atomic force microscopy.

Types and specifications	NSG03/Au
Materials	Single crystal silicon, N-type, 0.01-0.0025 Ohm cm, Antimony doped
Chip size	4.14 ×1.6 ×0.3 mm
Reflective side	Au
Conductive coating	Au (35nm), Ti adhesion layer (25A)
Cantilever number	1 rectangular
Tip curvature radius	~ 35 nm

Comment 2: Defects and nano-inclusion can significantly affect the property. To better understand the structure characteristics, it is better to briefly introduce the filtering process from Figure 4h to Figure 4i and highlight the related importance.

Author reply: In the revised manuscript, **Fig. 4h** presents a high-resolution transmission electron microscopy (HRTEM) image of the skutterudite films. As the nanostructures in the image are smaller than 10 nm, they may not be very distinct in the original gray scale image. To improve the visibility of these nanostructures, pseudo-color processing was applied using temperature table adjustments in the Digital Micrograph software. **Fig. 4i** displays the resulting pseudo-colored HRTEM image, which enhances the clarity of the nanostructures. A comparison of the images before and after pseudo-color

processing is illustrated in Fig. R3.

Fig. R2. The software operation page related to the pseudo-color mode.

Fig. R3. (Fig. 4h-i in the revised manuscript) The comparison of images before and after pseudo-color processing.

Comment 3: Please provide further discussions on strategies for enhancing ZT and the application

potentials in the sensor.

Author reply: The strategies for enhancing ZT are further explored, focusing primarily on electric and thermal transport aspects. The electric transport performance of flexible skutterudite films is optimized through composition control. Initially, DFT calculations aid in designing the band structure of the nominal component, guiding the selection of doping elements and their concentrations in experiments. The actual doping level of Ce significantly impacts carrier concentration and mobility, thereby optimizing electrical conductivity improvements^{1,6,9,10}. In this study, achieving high ZT values crucially hinges on the low thermal conductivity of the skutterudite film. The presence of grains within 100 nanometers in flexible skutterudite films exceeds 80%, effectively scattering phonons. Considering that the mean free path of phonons in CoSb_3 is approximately 80 nm, this scattering mechanism is pivotal in reducing thermal conductivity^{11,12}. The size of these nano-grains is smaller than the mean free path of phonons, leading to robust phonon scattering, thereby impeding the rise in thermal conductivity.

The flexible skutterudite thermoelectric film enables thermoelectric detection across a broad temperature spectrum, from room temperature to medium-to-high levels. Its outstanding features, including high temperature resistance, flexibility, and lightweight nature, make it suitable for applications in environments with complex shapes, particularly in medium and high temperature settings. Fire-resistant clothing plays a crucial role in safeguarding firefighters during firefighting endeavors. Leveraging the flexible thermoelectric film, a self-powered intelligent fire alarm system can be developed. This system can preemptively alert firefighters before their attire is compromised by fire, allowing them to take timely preventive measures to avoid burns^{13,14}. Utilizing thermoelectric film to enable precise temperature sensing and responsive fire alarm capabilities in combustible

materials holds significant importance for establishing a safe home environment ^{15,16}. Moreover, a basic thermoelectric hydrogen sensor can be created by applying a catalyst onto one side of the thermoelectric film. When exposed to a hydrogen-containing environment, the catalyst facilitates the reaction between hydrogen and oxygen, producing water vapor and releasing heat. Consequently, the end with the catalytic metal deposit becomes hot, while the end without it remains cooler. This temperature gradient generates an electrical signal through the Seebeck effect, enabling the detection of hydrogen concentration.

We have added this additional information on **Page 13** and **Page 15** of the revised manuscript.

Comment 4: There are minor grammatical errors that can be corrected.

Author reply: We have carefully checked the whole manuscript and revised the grammatical errors in the revised manuscript.

References

1. Tong, X. et al. Research progress of p-type Fe-based skutterudite thermoelectric materials. *Front. Mater. Sci.* **15**, 317-333 (2021).
2. Liu, Z.-Y., Zhu, J.-L., Tong, X., Niu, S. & Zhao, W.-Y. A review of CoSb₃-based skutterudite thermoelectric materials. *J. Adv. Ceram.* **9**, 647-673 (2020).
3. Gainza, J. et al. Unveiling the correlation between the crystalline structure of M-filled CoSb₃ (M = Y, K, Sr) skutterudites and their thermoelectric transport properties. *Adv. Funct. Mater.* **30**, 2001651 (2020).
4. Bae, S.H., Lee, K.H. & Choi, S.-M. Effective role of filling fraction control in p-type Ce_xFe₃CoSb₁₂ skutterudite thermoelectric materials. *Intermetallics* **105**, 44-47 (2019).
5. Son, G. et al. Control of electrical to thermal conductivity ratio for p-type La_xFe₃CoSb₁₂ thermoelectrics by using a melt-spinning process. *J. Alloys Compd.* **729**, 1209-1214 (2017).
6. Wan, S., Huang, X., Qiu, P., Bai, S. & Chen, L. The effect of short carbon fibers on the thermoelectric and mechanical properties of p-type CeFe₄Sb₁₂ skutterudite composites. *Mater. Design* **67**, 379-384 (2015).
7. Rogl, G. et al. New bulk p-type skutterudites DD_{0.7}Fe_{2.7}Co_{1.3}Sb_{12-x}X_x (X=Ge, Sn) reaching $ZT > 1.3$. *Acta Mater.* **91**, 227-238 (2015).
8. Tang, X.F., Chen, L.D., Goto, T., Hirai, T. & Yuan, R.Z. Synthesis and thermoelectric properties of p-type barium-filled skutterudite Ba_yFe_xCo_{4-x}Sb₁₂. *J. Mater. Res.* **17**, 2953-2959 (2002).
9. Liu, Z., Yang, T., Wang, Y., Xia, A. & Ma, L. Energy band and charge-carrier engineering in skutterudite thermoelectric materials. *Chin. Phys. B* **31**, 107303 (2022).
10. Tang, Y., Hanus, R., Chen, S.-w. & Snyder, G.J. Solubility design leading to high figure of

merit in low-cost Ce-CoSb₃ skutterudites. *Nat. Commun.* **6**, 7584 (2015).

11. Song, D.W. et al. Thermal conductivity of skutterudite thin films and superlattices. *Appl. Phys. Lett.* **77**, 3854-3856 (2000).
12. Savchuk, V., Boulouz, A., Chakraborty, S., Schumann, J. & Vinzelberg, H. Transport and structural properties of binary skutterudite CoSb₃ thin films grown by dc magnetron sputtering technique. *J. Appl. Phys.* **92**, 5319-5326 (2002).
13. He, H. et al. Temperature-arousing self-powered fire warning e-textile based on p–n segment coaxial aerogel fibers for active fire protection in firefighting clothing. *Nano-Micro Lett.* **15**, 226 (2023).
14. Jiang, C. et al. A high-thermopower ionic hydrogel for intelligent fire protection. *J. Mater. Chem. A* **10**, 21368-21378 (2022).
15. Zhao, Y. et al. Self-powered, durable and high fire-safety ionogel towards Internet of Things. *Nano Energy* **116**, 108785 (2023).
16. Li, G. et al. Thermoelectric and photoelectric dual modulated sensors for human internet of things application in accurate fire recognition and warning. *Adv. Funct. Mater.* **33**, 2303861 (2023).

REVIEWERS' COMMENTS

Reviewer #1 (Remarks to the Author):

It can be published as is.

Reviewer #2 (Remarks to the Author):

In the revision, my comments have been addressed appropriately. Especially, the strategies for enhancing ZT value of skutterudite films are constructive. I think that this paper can be recommended for publication in Nature Communications without further modifications.